# Action of Botulinum Neurotoxin E Type in Experimental Epilepsies

**DOI:** 10.3390/toxins15090550

**Published:** 2023-09-04

**Authors:** Flavia Antonucci, Yuri Bozzi

**Affiliations:** 1Department of Medical Biotechnology and Translational Medicine (BIOMETRA), University of Milan, via Fratelli Cervi 93, 20054 Milan, Italy; 2CNR Institute of Neuroscience, via Raoul Follereau 3, 20854 Vedano al Lambro, Italy; 3CIMeC—Center for Mind/Brain Sciences, University of Trento, Piazza della Manifattura 1, 38068 Rovereto, Italy; 4CNR Institute of Neuroscience, via Giuseppe Moruzzi 1, 56124 Pisa, Italy

**Keywords:** bacterial toxin, hippocampus, seizure, epilepsy

## Abstract

Botulinum neurotoxins (BoNTs) are zinc endopeptidases produced by the *Clostridium* genus of anerobic bacteria, largely known for their ability to cleave synaptic proteins, leading to neuromuscular paralysis. In the central nervous system, BoNTs are known to block the release of glutamate neurotransmitter, and for this reason, researchers explored the possible therapeutic action in disorders characterized by neuronal hyperactivity, such as epilepsy. Thus, using multidisciplinary approaches and models of experimental epilepsy, we investigated the pharmacological potential of BoNT/E serotype. In this review, written in memory of Prof. Matteo Caleo, a pioneer in these studies, we go back over the hypotheses and experimental approaches that led us to the conclusion that intrahippocampal administration of BoNT/E (i) displays anticonvulsant effects if prophylactically delivered in a model of acute generalized seizures; (ii) does not have any antiepileptogenic action after the induction of status epilepticus; (iii) reduces frequency of spontaneous seizures in a model of recurrent seizures if delivered during the chronic phase but in a transient manner. Indeed, the control on spontaneous seizures stops when BoNT/E effects are off (few days), thus limiting its pharmacological potential in humans.

## 1. Introduction: General Features and Experimental Applications of BoNT/s

Botulinum neurotoxins (BoNTs) are metalloproteases produced by Gram-positive bacteria of the *Clostridium* genus, whose neurotoxicity is widely acknowledged. They cleave specific synaptic proteins at neuromuscular junction making nerve communication impossible. Indeed, the persistent flaccid paralysis of variable extent caused by BoNT intoxication known as botulism is induced by the blockade of neurotransmitter release mainly at peripheral cholinergic nerve terminals of the skeletal and autonomic nervous system [1,2]. BoNT administration at neuromuscular junctions is currently practiced in clinical neurology and esthetic surgery in a series of conditions that require to deactivating muscular contraction [3,4].

BoNTs are traditionally classified into seven serotypes, designated with alphabetical letters from A to G [5]. However, next-generation sequencing techniques have led to the discovery of genes encoding for many novel BoNTs grouped within an existing serotype, even if characterized by different amino acid sequences [(see GeneBank and Uniprot databases at https://www.ncbi.nlm.nih.gov (accessed on 21 July 2023) and https://www.uniprot.org (accessed on accessed on 21 July 2023), respectively]. For this reason, new BoNTs are indicated with the letter of the serotype followed by a number (e.g., BoNT/A1, BoNT/A2…) [5,6,7]. Also, chimeric BoNTs resulting from naturally occurring recombination events within the *bont* genes have been identified. Chimeric BoNTs are labelled as the combination of two different serotypes as in the case of the chimeric BoNT/CD, where the light chain is derived from BoNT/C and the heavy chain from BoNT/D [8]. BoNT/H was isolated from a patient with infant botulism [9]. This toxin type is composed of a mosaic structure including regions of similarity to BoNT/A and F [10]; for this reason, BoNT/H is neutralized by antibodies against BoNT/A. Finally, BoNT-like sequences have been found in non-clostridial species such as *Weissella oryzae* and *Chryseobacterium piperi*, but the clinical implications or impacts of the BoNT-like toxins or sequences are not yet elucidated [11].

Regardless of their specific structural features, all BoNTs display similarities such as the presence of multiple domains that fulfil different functions during the intoxication process. The light chain (L-chain) includes the metalloprotease domain that specifically cleaves the SNARE (soluble N-ethylmaleimide-sensitive factor attachment protein receptor) proteins necessary for neurotransmitter exocytosis. The heavy chain (H-chain) N domain (the N terminus of the H chain) is essential for translocation of the L chain across the membrane of endocytic vesicles into the neuronal cytosol. Finally, the HC domain (the C terminus of the H chain) is responsible for presynaptic binding and endocytosis [6]. All BoNTs display common attractive pharmacological properties such as high potency and specificity, low diffusion, and persistent but reversible inhibition of neurotransmitter release, which make them unique drugs. Indeed, their use in clinics is growing more and more in conditions under which neurotransmitter release blockade is the only way to counteract hyperactivity of cholinergic nerve terminals. For instance, BoNT/A delivery is a widespread approach in both human therapies and cosmetic procedures. Since the ability to switch off neuronal communication is a common feature of all BoNTs, other serotypes have been also exploited. As an example, the use of BoNT/B has been investigated in preclinical studies upon injection in both the peripheral (PNS) and central (CNS) nervous system, and is now commercially available (Neurobloc, Elan Pharmaceuticals). Indeed, when the BoNTs are injected into the CNS, their potency and duration of action are comparable to those observed in the peripheral nervous system [12,13,14]. Importantly, in the CNS, they also retain the specificity for synaptic targets. Indeed, BoNT/B, /D, /F, and /G cleave only synaptic vesicle-associated membrane protein (VAMP) at single sites; BoNT/A and /E cleave only 25 kDa synaptosomal-associated protein (SNAP-25), whereas BoNT/C cleaves both SNAP-25 and syntaxin [15,16]. Since VAMP, SNAP-25 and syntaxin belong to the SNARE complex (the minimal machinery required for synaptic vesicle exocytosis also expressed in the CNS), it has been concluded that BoNT/s block neurotransmitter release in central neurons both in vitro and in vivo [17,18,19,20,21,22,23,24]. In the CNS, BoNTs prevent the release of the most important neurotransmitters and neuromodulators including glutamate, GABA, glycine, acetylcholine, and noradrenaline [17,18]; accordingly, altered cognitive and motor behaviors appear in animals injected with BoNTs in the brain [25,26]. BoNTs also inhibit the release of pain-modulating neurotransmitters such as substance P (SP) and calcitonin gene-related peptide (CGRP) by impairing synaptic vesicle fusion and modulating the transient receptor potential (TRP) of pain-sensing transmembrane receptors at the neuronal plasma membrane [27]. For this reason, the therapeutic use of BoNTs in reducing pain has received full attention. Importantly, BoNT delivery may also interfere with the secretion of pathological aggregates [28,29] as well as with the process of glutamate receptors insertion/trafficking during induction of synaptic activity and plasticity [30,31]. Indeed, the observation that the transsynaptic spreading of α-synuclein aggregates injected into the mouse striatum significantly decreases in the contralateral hemisphere of BoNT/B-treated animals led to the postulation of a novel hypothesis on the mechanisms of α-synuclein propagation [28]. Accordingly, in the context of a different neurodegenerative disorder (Alzheimer’s disease), it has been shown that tau release from rodent and human synaptosomes is a calcium- and SNAP-25-dependent mechanism sensitive to BoNT/A treatment. Indeed, the cleavage of SNAP-25 by BoNT/A modulates the release of pathologic tau [29,32]. Moreover, since the expression of SNARE fusion machinery has been largely assessed at the postsynaptic site, the delivery of BoNT/s has been exploited to better investigate AMPA and NMDA glutamate receptor insertion and diffusion during neuronal activity and plasticity. Both SNAP-23 and SNAP-25 have been suggested to play specific roles in the trafficking of NMDARs to synapses [33,34] and syntaxin-4 to regulate AMPA exocytosis in dendritic spines [30]. Syntaxin-3 and SNAP-47 are required for regulated AMPAR exocytosis during LTP, but not for constitutive basal AMPAR exocytosis, which depends on the R-SNARE protein synaptobrevin-2/VAMP2 [31]. In keeping with these findings, cleavage of SNAP-25 by BoNT/A in hippocampal slices in vitro abolished the ability of low-frequency synaptic stimulation to induce LTD at Schaffer collateral presynaptic release sites [35].

Taken together, these studies show that BoNTs may also serve as potent tools to study pathological processes in neurodegenerative diseases and to identify novel targets for therapeutic interventions in neurological conditions.

## 2. Effects of Intracerebral Administration of BoNT/E

### 2.1. In Vivo BoNT/E Effects upon Intrahippocampal Injection: Hypothesizing the Pharmacological Potential

Starting from the assumption that BoNTs act at CNS synapses and taking into consideration the positive pharmacological properties of BoNTs in terms of specificity, potency, and reversable effects, many researchers have explored the use of BoNTs in neurological conditions. In particular, in a study published in 2005 [36], we found that a single intrahippocampal infusion of BoNT/E, which selectively cleaves SNAP-25, results in a dramatic reduction in glutamate release. In vivo electrophysiological recordings from CA1 and CA3 pyramidal neurons indicated that spontaneous discharges were potently inhibited by BoNT/E treatment whereas immunodetection of cleaved SNAP-25 and loss of intact SNAP-25 lasted for at least three weeks. BoNT/E-injected rats showed significant deficits in acquisition of spatial learning in the Morris water maze test (MWM, a well-established hippocampus-dependent behavioral task), which were no longer detected when BoNT/E effects wore off, i.e., 5 weeks after in vivo delivery. We therefore hypothesized that BoNT/E might transiently silence neural activity in the CNS and for this reason could be exploitable in pathological conditions characterized by brain hyperactivity. In this context, the discovery that glutamatergic and GABAergic neurons are differentially sensitive to BoNT/E due to the preferential expression of the BoNT/E target (SNAP-25) at glutamatergic synapses [37] prompted us to further assess this issue. We hypothesized that the silencing of spontaneous spiking activity induced by BoNT/E administration could be beneficial in hyperactivity disorders such as epilepsy.

### 2.2. Acute Anticonvulsant Effects of BoNT/E

Epilepsy is a neurological condition largely diffused in the human population, characterized by the occurrence of spontaneous recurrent seizures responsible for hippocampal neuronal loss, reduced cognitive functions, and psychiatric comorbidities [38]. It presents as an exceptionally multifaceted cluster of diseases that vary in etiology, age of onset, type of seizure, and neurological and neuropathological manifestations. However, human genetic studies yielded a steady trickle of discoveries, thanks to which the critical role of ion channels in epilepsy has been postulated [39]. According to this idea, changes in ion channel functionality cause massive depolarization of neurons leading to excessive glutamate release responsible for seizure generation and brain damage [40,41]. Thus, drugs able to affect abnormal glutamate release by interfering with neurotransmitter release machinery may produce anti-ictal and neuroprotective effects. Starting from this concept, we investigated BoNT/E effects in the epilepsy context. Regarding this issue, we exploited a model of acute electroencephalographic (EEG) seizures triggered by intrahippocampal injection of kainic acid (KA; Figure 1) [42,43]. Administration of KA in rodents is widely used to model human temporal lobe epilepsy (TLE), a common form of epilepsy characterized by recurrent seizures originating in the temporal lobe (typically in the hippocampus) and often accompanied by hippocampal neurodegeneration [44]. The effects depend on the species (rat vs. mouse) and route of administration (intrahippocampal vs. systemic). In our studies, we used different KA models of TLE, which will be described in the following paragraphs.

In our first study, adult rats received a single intracerebral BoNT/E (or vehicle) application, and two days later, seizures were induced via focal KA delivery to the hippocampus. Intrahippocampal administration of KA in rats provokes acute hippocampal seizures [42,43] and is generally used to model focal hippocampal seizures that occur in human TLE and to test potential anticonvulsant therapies. Analysis of EEG recordings allowed for detection of the expected ictal activity in vehicle-treated animals, and quantification of seizure activity showed clear anti-ictal effects of the neurotoxin; BoNT/E was significantly more effective than the common antiepileptic drug phenytoin in reducing KA-induced seizures [36].

We further confirmed the anti-ictal effects of BoNT/E using additional experiments performed on behavioral seizures induced through systemic administration of KA [45,46]. Different from the intrahippocampal administration model, systemic KA administration in rats reliably induces acute seizure, followed by the development of cognitive deficits and the occurrence of massive hippocampal neurodegeneration, thus allowing for the modelling of TLE in a more comprehensive way [45,46]. In this case, rats received intrahippocampal infusions of BoNT/E (or vehicle) and, one day later, a single intraperitoneal injection of a convulsive dose of KA. KA treatment had a similar convulsant effect in both naive and vehicle-injected animals, as indicated by the visual inspection of seizure progression. Rats showed initial subconvulsant behaviors (immobility and wet dog shake movements) culminating in generalized tonic–clonic motor seizures with rearing and falling according to Racine’s classification [47]. Importantly, progression of clinical signs was dramatically different in BoNT/E-injected animals: the maximum seizure score assigned to each experimental animal clearly displayed the highly significant anticonvulsant effect of BoNT/E. In parallel, activity mapping studies with *c-fos* mRNA in situ hybridization experiments revealed strong bilateral activation in the hippocampus, thalamus, and cerebral cortex of animals treated with KA and a dramatic decrease in *c-fos* induction in BoNT/E-treated rats [36].

Finally, since seizure activity induced through systemic KA is responsible for the occurrence of cognitive defects in animals [48,49], we explored the possibility that the anti-ictal activity of BoNT/E could also result in a protective phenotype. We tested cognition in the MWM; since BoNT/E-injected animals displayed impaired but reversable cognitive functions due to the transient blockade of neurotransmitter release and synaptic activity, we tested animals treated with BoNT/E (or vehicle) and systemic KA five weeks after BoNT/E delivery, i.e., at the end of BoNT/E effects. Spatial learning performances in rats treated with BoNT/E + KA rats were significantly superior to those of control rats treated with vehicle + KA, indicating that BoNT/E was able to prevent cognitive deficits induced by KA. Also, immunostaining for the neuronal marker NeuN used to evaluate the extent of neuronal loss in CA1, CA3, and hilus of hippocampus revealed a significant neuroprotection in the dorsal hippocampus of BoNT/E + KA-treated animals in contrast to the severe lesions found in CA1 and CA3 of vehicle + KA-treated rats [36]. From a molecular point of view, BoNT/E-mediated neuroprotection is achieved by preventing the upregulation of apoptotic proteins such as phosphorylated c-Jun and cleaved caspase 3, which occurs in hippocampal neurons following KA seizures [50].

These results clearly supported the notion that intrahippocampal delivery of BoNT/E produced anti-ictal effects in rodents but did not lead to any conclusions regarding the possible antiepileptogenic action of the neurotoxin. Indeed, to address this point, we exploited the rapid electrical kindling of the ventral hippocampus. According to this model, rapid kindling was induced in the hippocampus by delivering constant current stimuli through a bipolar electrode [51,52] and behavior observed and scored according to Racine’s classification [47]. By measuring the behavioral progression of seizures during kindling and the duration of primary and secondary afterdischarge during kindling, we concluded that BoNT/E delayed hippocampal kindling rate, thus turning out to be an anticonvulsant and potentially antiepileptogenic drug [36].

**Figure 1 toxins-15-00550-f001:**
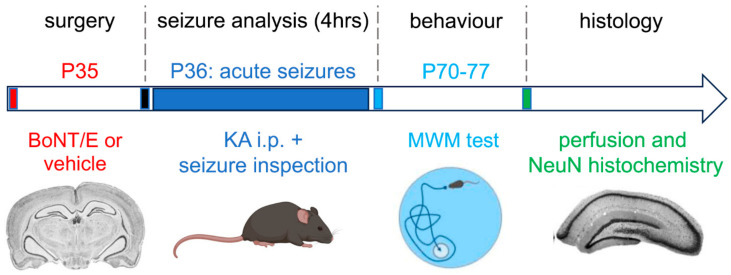
Experimental approach to investigating the anticonvulsant effects of BoNT/E. In our first study, BoNT/E was administered before the induction of KA seizures in rats, revealing both anticonvulsant and neuroprotective effects [36,50]. Abbreviations: KA: kainic acid; MWM: Morris water maze test. The mouse and MWM sketches were generated using biorender.com.

### 2.3. Evaluation of BoNT/E Antiepileptogenic Effects

The abovementioned results indicated that BoNT/E might be used as a novel tool to interfere with the mechanisms underlying seizure generation. Then, it was important to clearly assess the possible antiepileptogenic effects of BoNT/E in a model of chronic epilepsy. Thus, in a second study (Figure 2), we tested whether delivery of BoNT/E was effective in preventing limbic epileptogenesis following an episode of status epilepticus (SE). To this purpose, we exploited another well-characterized model of TLE, where epileptogenesis was triggered by intrahippocampal injection of KA in adult mice [53,54]. Differently from what was observed in the rat, intrahippocampal injection of KA in mice consistently induces a form of chronic epilepsy resembling human TLE: spontaneous recurrent seizures appear in the hippocampus and temporal lobe several days after KA administration. For this reason, we preferred the mouse model to the rat one for investigating the antiepileptogenic effects of BoNT/E (i.e., the capability of BoNT/E to halt the development of chronic seizures). In this model, chronic recurrent seizures appear after a latent period of about 14 days, and a lesion resembling hippocampal sclerosis (often observed in human TLE as well) progressively develops in the injected hippocampus [53,54]. In addition, chronic seizures induced by intrahippocampal KA in mice are accompanied by a severe hippocampal neuropathology resembling that observed in cases of human TLE; histopathological findings include the sprouting of hippocampal mossy fibers (granule cell axons) into the dentate gyrus and CA3 layer, as well as granule cell dispersion [55]. After characterizing the effects of a single BoNT/E infusion into the mouse hippocampus (i.e., the expected cleavage of SNAP-25, the tuning of hippocampal cell firing, and the BoNT/E duration of action), we tested whether the administration of BoNT/E into the KA-treated hippocampus interfered with epileptogenesis. Using EEG recordings, we found that KA-treated mice injected with BoNT/E three hours after KA infusion displayed delayed epileptogenesis. Indeed, the first electrographic seizure occurred after a longer latency time, indicating that epileptogenesis was significantly prolonged in KA-treated mice injected with BoNT/E [56]. However, all toxin-treated and control animals eventually developed the characteristic features of focal chronic epilepsy. Unexpectedly, histological analysis carried out to evaluate neuronal loss in hippocampus and granular cell dispersion in the hilus of dentate gyrus revealed that BoNT/E afforded significant neuroprotection in CA1 area after SE and reduced dispersion of dentate granule cells [56]. In addition, we showed that KA-treated mice injected with BoNT/E displayed a decreased expression of reelin mRNA [56], an extracellular matrix protein that controls neuronal migration and whose expression correlates with the extent of granule cell dispersion [57].

No prevention of neuronal loss in CA3 was found in BoNT/E-treated rodents, probably due to the direct interaction of KA with high affinity postsynaptic kainate receptors in the CA3 hippocampal subfield. In this case, the blockade of synaptic transmission with BoNT/E did not produce any survival action. Finally, we evaluated the sprouting of mossy fibers using NPY immunohistochemistry [58,59] and found a robust increase in NPY immunoreactivity in dentate granule cells, the inner molecular layer, and mossy fiber pathway of all KA mice, regardless of the treatment condition [55].

These results indicate that administration of BoNT/E produced positive effects in terms of neuroprotection and granule cell dispersion following SE induced by intrahippocampal KA in mice. BoNT/E also delayed the appearance of spontaneous paroxysms without any beneficial effect on seizure frequency during the chronic phase. Thus, interfering with synaptic transmission partially protects against certain histopathological changes but does not halt the development of chronic epilepsy in this mouse model of TLE.

### 2.4. Evaluation of BoNT/E Effects on Chronic Seizures

Finally, we decided to test whether unilateral delivery of BoNT/E into the hippocampus of chronically epileptic mice is effective in reducing spontaneous recurrent seizures (SRS; Figure 3) [60].

To this end, mice received a single infusion of KA and were implanted with bipolar electrodes in the dorsal hippocampus to allow for chronic EEG recordings. As described before, this procedure reliably induces chronic SRS after a latent period of two–three weeks [53,54,55]. Three weeks after KA-induced status epilepticus, animals were injected with BoNT/E (or vehicle) into the hippocampus via the guide cannula and SRS-monitored for five consecutive days. Quantification of the EEG data revealed a significant reduction in SRS frequency in BoNT/E mice, and total time spent in ictal activity was also significantly decreased by BoNT/E for at least 5 days. Unfortunately, the effects of BoNT/E were transient, as mice recorded 21 days after treatment (i.e., at the end of toxin effects) showed a return to the baseline, pre-drug level of seizures. Delivery of BoNT/E during the chronic phase was ineffective in reducing both hippocampal sclerosis and granule cell dispersion, as evaluated by analysis of Nissl-stained dorsal hippocampal sections [60].

In conclusion, these last experiments indicated that BoNT/E might reduce the incidence of chronic seizures, but BoNT/E effects were transient, disappearing at the end of treatment.

## 3. Discussion

Our studies, summarized in Table 1, explored the innovative and unexplored topic that BoNT/E may be beneficial in epilepsy. That BoNTs affected both peripheral and central synapses was already a fact at the end of last century but, since BoNTs do not cross the blood–brain barrier, it was hard to test their therapeutic potential against neurological conditions. The first demonstration of the in vivo central toxicity and recovery of health status after intracerebroventricular (i.c.v.) injection of BoNT/A or /B was assessed only in 2003 [11] followed by the clear proof that BoNTs’ delivery in the CNS produced cognitive dysfunctions [26].

At that time, evidence for the central effect of BoNTs upon peripheral administration was also largely described but ascribed to indirect mechanisms of peripheral alteration of central sensorimotor integration [61,62]. In 1995, driven by the evidence that BoNT/A reduces the amplitude and frequency of muscle contractions in blepharospasm and hemifacial spasm [63], electromyographic tremor bursts [64], and electromyographic burst discharges in one case of spinal myoclonus [65], Tarsy and Schachter evaluated the effects of injections of BoNT/A in biceps and triceps muscles of patients with epilepsia partialis continua (EPC) [66]. However, no positive results were observed in two patients with EPC when treated with BoNT/A [66] at doses that equaled or exceeded those used in upper extremity tremor or dystonia [64]. This was likely due to the peripheral instead of central infusion of the toxin. Thus, our investigation regarding in vivo BoNT/E effects in epilepsy was extremely ambitious and of potentially high scientific and clinical impact. Indeed, our studies not only explored effects of BoNT/E in different rodent models of acute and chronic seizures but also showed a dissociation between specific histopathological changes and mechanisms that support epileptogenesis. Our initial results regarding the anticonvulsant effects of BoNT/E were the “proof of concept” that the starting hypothesis was correct: drugs able to interfere with neurotransmitter release machinery such as BoNT/E may produce anti-ictal and neuroprotective effects. From a pharmacological point of view, it was crucial to assess the possible antiepileptic effect of BoNT/E. Results clearly confirmed our idea that the blockade of glutamate release is also sufficient to reduce seizure frequency in a chronic epileptic condition, but due to the reversibility of BoNT/E activity, the effect on spontaneous recurrent seizures was confined to the duration of BoNT/E action (i.e., a couple of weeks). Of course, this represents a limiting factor because it means that repetitive BoNT/E injections in the CNS would be required to preserve antiepileptic effects. Thus, our data indicate that BoNT/E is, in theory, useful in hyperexcitability conditions but almost unusable due to its short duration of action. To overcome this point, we also explored the therapeutic potential of BoNT/A in epilepsy, since a single BoNT/A injection into the mouse hippocampus resulted in a longer-lasting blockade of glutamate release due to the production of a shorter SNAP-25 fragment. Unfortunately, appearance of BoNT/A-truncated SNAP-25 fragments persisted up to 6–9 months [67], making the evaluation of BoNT/A effects in the model of chronic epilepsy impossible. Of note, since BoNTs can be easily engineered and produced using recombinant methods, we cannot exclude that, in the future, new technological approaches able to target BoNT/E directly to the epileptic focus upon systemic/peripheral administrations may lead to new pharmacological BoNT/E-based approaches in epilepsy. Indeed, several laboratories are working in this direction, generating new BoNT/s chimeras to modify/prolong a specific BoNT serotype activity [68] to reduce exocytosis from nonneuronal cells [69], or to increase potency over the wild type [70]. As an example, the generation of a photoactivatable BoNT/B (an engineered toxin serotype activable by blue light) showed for the first time that optogenetic tools may be used to disrupt excitatory neurotransmission, achieving persistent synaptic inhibition [71].

Finally, our data showed that BoNT/E does not display an antiepileptogenic action. Delivery of BoNT/E three hours after SE prolonged epileptogenesis and reduced the extent of hippocampal lesions but not the frequency of spontaneous seizures. This means that epileptogenic mechanisms responsible for spontaneous seizures are not driven by an excessive hippocampal glutamate release, which is instead necessary for specific anatomical modifications during epileptogenesis.

Another possible explanation for our results is that in our model, even if KA delivery occurs focally in the dorsal hippocampus, additional epileptic foci may contribute to the establishment of recurrent spontaneous seizures. Since BoNT/E effects remain confined to the injection site, a single BoNT/E shot is not enough to reduce the high glutamate release produced by distant hyperactive neuronal populations. Up to now, effects of multiple toxin injections in the KA focal model have not been assessed.

Anatomically, BoNT/E delivery prevents the loss of CA1 neurons and dispersion of dentate granular cells but does not interfere with CA3 neuronal death and with another histopathological marker of chronic seizures, mossy fiber sprouting. Whereas BoNT/E-mediated neuroprotection may be associated to the inhibition of the caspase-3 pathway in KA-injured hippocampal neurons [50], action on dispersion of granular cells is mainly ascribable to upregulation of reelin mRNA. Indeed, in both humans and experimental models of TLE, the significant downregulation of reelin expression has been defined as a major determinant of granule cell dispersion [57,72,73]. Thus, loss of CA3 neurons (probably induced by the direct activation of postsynaptic kainate receptors) and mossy fiber sprouting, not impacted by BoNT/E delivery, may be responsible for epileptogenesis in this model.

## 4. Conclusions

Our studies revealed specific features of BoNT/E upon in vivo delivery in the CNS in both physiological and pathological conditions. In addition, we described a different ability of BoNT/E to interfere with neuronal hyperactivity and identified a dissociation between specific histopathological changes and epileptogenesis and chronic epilepsy.

## Figures and Tables

**Figure 2 toxins-15-00550-f002:**
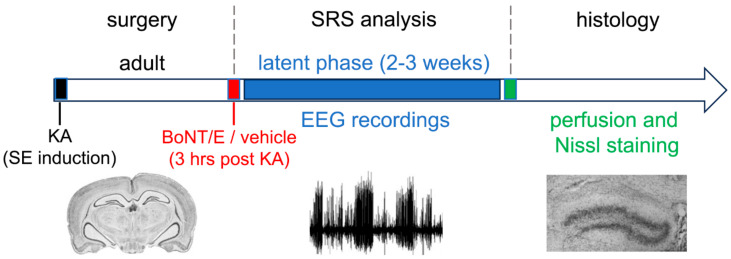
Experimental approach to investigating the antiepileptogenic effects of BoNT/E. In a second study, we administered BoNT/E 3 h after intrahippocampal KA. In this experimental paradigm, BoNT/E showed antiepileptogenic and neuroprotective effects [56]. Abbreviations: KA: kainic acid; SE: status epilepticus; SRS: spontaneous recurrent seizures.

**Figure 3 toxins-15-00550-f003:**
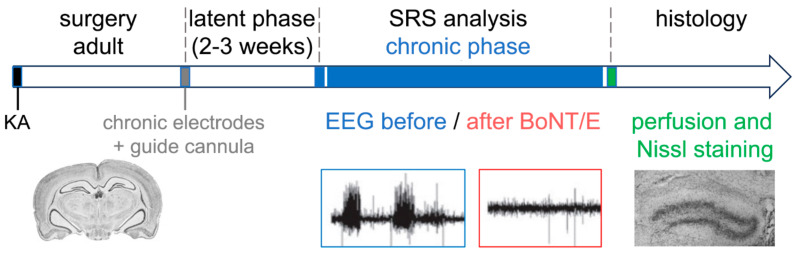
Experimental approach to investigating the antiepileptic effects of BoNT/E. BoNT/E administered three weeks after KA-induced SE in mice failed to permanently suppress chronic seizures and to exert a protective effect against hippocampal granule cell dispersion [60].

**Table 1 toxins-15-00550-t001:** Action of BoNT/E in rodent models of experimental epilepsies.

Animal Model	Bont/E Delivery	BoNT/E Effects	Reference
Intrahippocampal KA (rat)	Intrahippocampal, before KA	Reduced EEG seizures (more effective than phenytoin).	[36]
Systemic KA (rat)	Intrahippocampal, before KA	Prevented KA-induced spatial learning deficits (Morris water maze). Protection against KA-induced hippocampal cell loss.	[36]
Hippocampal kindling (rat)	Intrahippocampal, before KA	Delayed kindling.	[36]
Intrahippocampal KA (rat)	Intrahippocampal, before KA	Prevented the upregulation of phosphorylated c-Jun and cleaved caspase 3.	[50]
Intrahippocampal KA (mouse)	Intrahippocampal, 3 h after KA	Slightly delayed (but not prevented) epileptogenesis. Protection against hippocampal cell loss and dentate granule cell dispersion.	[56]
Intrahippocampal KA (mouse)	Intrahippocampal, 21 days after KA	Transient (5 days) but not long-lasting reduction in SRS. No reduction of both hippocampal sclerosis and granule cell dispersion.	[60]

## Data Availability

No new data were created for preparing this review. Already published original data are available upon request.

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
