# Peer review of "Action of Botulinum Neurotoxin E Type in Experimental Epilepsies"

_toxins, 2023, doi:10.3390/toxins15090550_

Round 1
Author Response
- The mechanism of epilepsy should be described in detail. Since it is difficult to understand from the manuscript what kind of state epilepsy is in each model, you should describe in detail.
Response. We thank the Reviewer for this request, which allowed us to better clarify experimental models of epilepsy mentioned in our review. We addressed this issue in chapters 2.2 and 2.3 and 2.4.
- Since the titles (2.2~2.4) were similar, I thought it would be better to clarify.
Response. The titles have been changed to “2.2 Acute anticonvulsant effects of BoNT/E” and “2.4 Evaluation of BoNT/E effects on chronic seizures”.
- zinco-endopeptidase→zinc-endopeptidase
Response. The word has been corrected in the Abstract.
Reviewer 2 Report
The article is very well written and represents a very well-crafted scientific review, with research references dating from 1972 to 2023.
However, some small details need to be changed for the article to be accepted:
1). Missing Keywords and Key Contribution. Please insert.
two). The figures, despite being very didactic, deviate from the standard of the manuscript. I recommend following the same font size and font (arial, palatine, times) used in writing the manuscript. The way they are, they become strange and lose focus/academic quality. Avoid using Capslock words in the figures.
3). In my pdf file Table 1 is superimposed on the text of the article. Please correct. I don't know if this effect of formatting error occurred in the word version, in the pdf yes (line 271). What is the difference between rat and mouse?
4). The Acknowledgments item is incomplete (line 341) and needs to be written verbatim.
5). Author contribution. I believe that there are items in the Toxins rules that should be added to this item. Please also rewrite. For example, conceptualization, writing, formatting, proofreading....
6). References are not standardized, as per Toxins rules. Please review one by one and format. For example, in article references, the year must be in bold, the name of the journal in italics, and most importantly, articles that have a DOI must be inserted. Therefore, review, format and complete all references.
After these minor corrections, the article can be accepted.
Congratulations to the authors for the excellent review article.
Sincerely
Author Response
1) Missing Keywords and Key Contribution. Please insert.
Response. Keywords and Key Contribution have been added.
2) The figures, despite being very didactic, deviate from the standard of the manuscript. I recommend following the same font size and font (arial, palatine, times) used in writing the manuscript. The way they are, they become strange and lose focus/academic quality. Avoid using Capslock words in the figures.
Response. We modified the figures according to the journal style and Reviewer’s suggestion. Some words have been maintained in capital letters since they are acronyms (e.g. KA, EEG, SRS…).
3) In my pdf file Table 1 is superimposed on the text of the article. Please correct. I don't know if this effect of formatting error occurred in the word version, in the pdf yes (line 271). What is the difference between rat and mouse?
Response. The superimposition of Table 1 on the text was likely due to wrong formatting during pdf conversion of the manuscript. For publication purposes, we are ready to provide the manuscript in word format, as well as the text, figures, and table in separate files.
The major difference between the rat (refs. 36 and 49) and mouse (refs. 55 and 59) models relates to the different effects of intrahippocampal administration of KA in the two species. Intrahippocampal KA in rats induces acute seizures, and hippocampal cell loss, but chronic seizures do not generally appear in this model. Conversely, intrahippocampal administration of KA consistently induces a severe hippocampal neuropathology accompanied by chronic seizures. For this reason, the mouse model has been preferred to the rat one to investigate the effect of BoNT/E on the development and termination of chronic seizures. This is now specified in the revised version (chapter 2.3).
4) The Acknowledgments item is incomplete (line 341) and needs to be written verbatim.
Response. Acknowledgements have been corrected according to Toxins guidelines.
5) Author contribution. I believe that there are items in the Toxins rules that should be added to this item. Response. The author contribution section has been corrected according to Toxins guidelines.
6) References are not standardized, as per Toxins rules. Please review one by one and format. For example, in article references, the year must be in bold, the name of the journal in italics, and most importantly, articles that have a DOI must be inserted. Therefore, review, format and complete all references.
Response. References have been formatted according to the journal style and Reviewer’s suggestions.
Reviewer 3 Report
This review explores the potential therapeutic application of Botulinum neurotoxins (BoNTs) in disorders characterized by neuronal hyperactivity, such as epilepsy. BoNTs are zinco-endopeptidases produced by anaerobic bacteria, known for their ability to induce neuromuscular paralysis by cleaving synaptic proteins. In the central nervous system, BoNTs can block the release of the glutamate neurotransmitter, making them intriguing candidates for epilepsy treatment.
The review focuses on the BoNT/E serotype and investigates its pharmacological potential through various experimental approaches and models of epilepsy. The findings suggest that intrahippocampal administration of BoNT/E can display anticonvulsant effects when delivered prophylactically in a model of acute generalized seizures. However, it does not have any antiepileptogenic action after the induction of status epilepticus. Additionally, it can reduce the frequency of spontaneous seizures in a model of recurrent seizures if delivered during the chronic phase, but its effects are transient, lasting only a few days. This limitation restricts its long-term pharmacological potential in humans.
The review pays tribute to the late Prof. Matteo Caleo, who pioneered these studies and highlights the multidisciplinary approaches used to investigate BoNT/E's therapeutic potential in epilepsy treatment.
However, there is one thing that should be addressed before the paper being accepted.
“They cleave specific synaptic proteins at neuromuscular junction making the nerve communication impossible. Indeed, the persistent flaccid paralysis of variable extent caused by BoNTs intoxication known as botulism is induced by the blockade of neurotransmitter release mainly at peripheral cholinergic nerve terminals of the skeletal and autonomic nervous system [1,2].”
After this sentence, please note and add that botulinum neurotoxin is clinically practiced in neuromuscular junctions for deactivating muscular contractions in aesthetics and neurology.
Refer “Sihler's staining technique: How to and guidance for botulinum neurotoxin injection in human muscles” and “Scientific review of the aesthetic uses of botulinum toxin type A”
Author Response
- “They cleave specific synaptic proteins at neuromuscular junction making the nerve communication impossible. Indeed, the persistent flaccid paralysis of variable extent caused by BoNTs intoxication known as botulism is induced by the blockade of neurotransmitter release mainly at peripheral cholinergic nerve terminals of the skeletal and autonomic nervous system [1,2].” After this sentence, please note and add that botulinum neurotoxin is clinically practiced in neuromuscular junctions for deactivating muscular contractions in aesthetics and neurology. Refer “Sihler's staining technique: How to and guidance for botulinum neurotoxin injection in human muscles” and “Scientific review of the aesthetic uses of botulinum toxin type A”
Response. We thank the Reviewer for this important suggestion. We added the sentence as requested, along with the suggested references. The numbering of all references has been changed accordingly.
Round 2
Reviewer 1 Report
I encourage its publishing the revised paper in "Toxins".